# Estimating Uncertainty in PET Image Reconstruction via Deep Posterior Sampling

**Tin Vlašić**                                                TIN.VLASIC@FER.HR

**Tomislav Matulić**                                  TOMISLAV.MATULIC@FER.HR

**Damir Seršić**                                            DAMIR.SERSIC@FER.HR

*Faculty of Electrical Engineering and Computing, University of Zagreb, Zagreb, Croatia*

**Editors:** Accepted for publication at MIDL 2023

## Abstract

Positron emission tomography (PET) is an important functional medical imaging technique often used in the evaluation of certain brain disorders, whose reconstruction problem is ill-posed. The vast majority of reconstruction methods in PET imaging, both iterative and deep learning, return a single estimate without quantifying the associated uncertainty. Due to ill-posedness and noise, a single solution can be misleading or inaccurate. Thus, providing a measure of uncertainty in PET image reconstruction can help medical practitioners in making critical decisions. This paper proposes a deep learning-based method for uncertainty quantification in PET image reconstruction via posterior sampling. The method is based on training a conditional generative adversarial network whose generator approximates sampling from the posterior in Bayesian inversion. The generator is conditioned on reconstruction from a low-dose PET scan obtained by a conventional reconstruction method and a high-quality magnetic resonance image and learned to estimate a corresponding standard-dose PET scan reconstruction. We show that the proposed model generates high-quality posterior samples and yields physically-meaningful uncertainty estimates.

**Keywords:** Bayesian inference, conditional generative adversarial network, deep generative model, inverse problem, positron emission tomography, uncertainty quantification.

## 1. Introduction

In inverse problems, the goal is to reconstruct an unknown signal, image or shape from a set of observations obtained by a forward process, which is typically non-invertible. Of particular interest are ill-posed inverse problems – reconstructing a *unique* solution that matches the observations is almost impossible unless there is some prior knowledge about the observed phenomenon. Recently, deep learning techniques demonstrated remarkable results in solving various inverse problems (Ongie et al., 2020), and are currently impacting the reconstruction methods. Learning-based approaches leverage large datasets in order to *i)* directly compute regularized reconstructions (Kulkarni et al., 2016), or *ii)* train deep generative models that regularize inverse problems by constraining their solutions to remain on a learned manifold (Bora et al., 2017; Vlašić et al., 2022).

In general, deep learning methods reconstruct a single solution. Since there are many plausible solutions that match the observations to within the noise level, in highly ill-posed and noise-corrupted problems, a single solution can be misleading or inaccurate. There are many situations where critical decisions are based on the solution of an ill-posed inverse

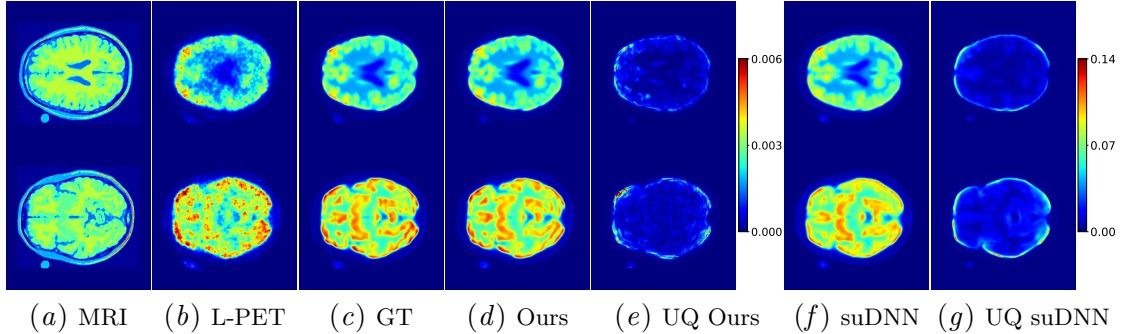

$(a)$ MRI    $(b)$ L-PET    $(c)$ GT    $(d)$ Ours    $(e)$ UQ Ours    $(f)$ suDNN    $(g)$ UQ suDNN

Figure 1: Standard-dose PET (S-PET) reconstructions and corresponding physically-meaningful measures of uncertainty (UQ) estimated using our proposed method from MRI and low-dose PET (L-PET) images. We compare our method with the suDNN framework reported by Sudarshan et al. (2021).

problem, especially in medicine (Begoli et al., 2019). In such cases, estimating uncertainty is key and leads to a more reliable interpretation of the reconstruction.

Bayesian inversion is a method that allows drawing conclusions from the observed measurements using a statistical framework. Its probabilistic characteristic leads to consistent uncertainty quantification (UQ) by a posterior probability distribution. Let $x$ and $y$ denote the unknown model parameters we seek and the observations, respectively, that are realizations of random vectors $X \in \mathcal{X}$ and $Y \in \mathcal{Y}$. In Bayesian inversion, the goal is to recover the posterior distribution $p_{X|Y}$, i.e., the conditional probability distribution of the model parameters given the observations which is expressed using the Bayes rule as

$$p_{X|Y}(x|y) = \frac{p_{Y|X}(y|x)p_X(x)}{\int_x p_{X,Y}(x,y)dx}, \tag{1}$$

where $p_{X,Y}$ is the joint distribution. In high-dimensional real-world inverse problems, computing posterior quickly becomes unfeasible due to the intractability of calculating $\int_x p_{X,Y}(x,y)dx$ and unavailability of an exact knowledge of $p_X$. One way of approximating the posterior is by using the variational Bayesian methods (Blei et al., 2017). In particular, deep learning-based variational methods have recently shown promising results for posterior approximation (Adler and Öktem, 2018; Khorashadizadeh et al., 2022b; Meng and Kabashima, 2022) and estimating uncertainties in the solutions of ill-posed inverse problems (Adler and Öktem, 2019; Abdar et al., 2021; Khorashadizadeh et al., 2022a).

Positron emission tomography (PET) is a medical imaging technique with a wide range of clinical applications in the evaluation of the pathophysiology of brain disorders, such as dementias, epilepsy, movement disorders, and brain tumors (Lameka et al., 2016). In contrast to magnetic resonance imaging (MRI) and computed tomography (CT), which are more suited for studying the anatomy, PET is referred to as a functional imaging technique that measures biological activity. In health care centers, PET is often simultaneously combined with CT or MRI, which results in hybrid scanners. Due to the physics and instrumentation,

PET image reconstruction is mainly low quality and low resolution, and CT and MRI reconstructions in the hybrid scanners are often employed to enhance it.

Prior to the acquisition, a radiopharmaceutical labeled with positron-emitting radioisotopes such as $^{11}$C and $^{18}$F is administered. A PET image is reconstructed from a set of observations obtained by $\gamma$-ray detectors that detect opposing pairs of photons produced in the annihilation event (Lameka et al., 2016), making it an ill-posed inverse problem. The radiation dose to the patient is an important aspect in PET imaging – its amount is positively correlated with the quality of PET image reconstruction. Ideally, one would like to reconstruct high-quality images with as small as possible amount of the radiation dose. With the recent emergence of deep learning methods, numerous data-driven approaches have been proposed for the enhancement of reconstructions from low-dose PET (L-PET) imaging (Reader et al., 2021; Pain et al., 2022). However, these methods return a point estimate without an associated measure of uncertainty. Since critical decisions are based on PET imaging reconstructions and PET is a highly sensitive imaging technique, it is of great importance to assess the uncertainty in the solution.

In this paper, we propose a framework for estimation of standard-dose PET (S-PET) image reconstruction with a corresponding UQ from L-PET and high-quality MRI images via sampling from the posterior. To obtain L-PET images from L-PET scanning, we use the maximum likelihood expectation maximization (MLEM) reconstruction algorithm. Our method achieves high-quality reconstructions with a physically-meaningful measure of uncertainty. Examples of reconstructions obtained by our method are given in Figure 1. The proposed method is based on a conditional generative adversarial network (cGAN) (Mirza and Osindero, 2014) whose generator is trained to output posterior samples. Our generator's architecture comprises of residual-in-residual dense blocks (RRDBs) proposed in (Wang et al., 2018a). The generator is conditioned on both the L-PET and MRI images and stochasticity is achieved by injecting per-pixel noise in every dense block of the network. Thus, the conditional input controls the global effects on the reconstruction and noise affects only stochastic variation. Our generator is able to produce a variety of plausible S-PET reconstructions which are consistent with the measurements given the same L-PET input.

## 2. Related Work

Currently, there are two main approaches using deep learning in PET image reconstruction. The first approach is the direct one, i.e., learning an encoding from the raw sinogram data to the desired S-PET image (Häggström et al., 2019; Hu et al., 2021). At the moment, the direct deep learning methods for PET image reconstruction look to be impractical, demanding huge amounts of computational memory and training data (Reader and Schramm, 2021).

The second approach is using deep learning methods for the enhancement of PET images obtained by the conventional reconstruction methods that are often simple and rapid, e.g., filtered backprojection (FBP). There are numerous papers proposing a myriad of different deep neural networks for this task, but two architectures prevail: U-Net (Ronneberger et al., 2015) and generative adversarial network (GAN) (Goodfellow et al., 2014). Chen et al. (2019) and Liu and Qi (2019) proposed U-Net-based methods and Garehdaghi et al. (2021) and Chen et al. (2021) proposed residual U-Net frameworks to predict S-PET images from ultra-L-PET (uL-PET) images in addition with corresponding MRI images. Sanaat et al.

(2020) used a U-Net for prediction of S-PET images and corresponding sinograms from low-dose counterparts. Wang et al. (2018b) proposed a patch-based 3D cGAN framework to estimate S-PET images from L-PET reconstructions. Lei et al. (2019) employed CycleGAN for a whole-body PET image estimation from L-PET scans. Ouyang et al. (2019) developed a cGAN framework with feature matching and task-specific perceptual loss for uL-PET image reconstruction. Jeong et al. (2021) used a cGAN framework with a U-Net-based generator for restoration of amyloid S-PET images from L-PET data. Wang et al. (2018c) used a U-Net-based generator in cGAN for S-PET estimation from a fusion of L-PET and multimodal MRI images. Luo et al. (2022) developed adaptive rectification-based GAN with spectrum constraint to synthesize S-PET images from L-PET ones. Finally, several papers proposed different convolutional neural network (CNN)-based supervised learning models for predicting S-PET images from L-PET ones (Xiang et al., 2017; Gong et al., 2018; Spuhler et al., 2020; Song et al., 2020).

While the aforementioned frameworks tend to restore S-PET images from L-PET reconstructions as well, in contrast to our framework, neither of them use the residual-in-residual CNN architecture, but typically the U-Net architecture for both supervised and unsupervised learning models. Moreover, they return a single solution of the problem without a measure of uncertainty, while our framework allows for the UQ. The most related work to our paper is a work from Sudarshan et al. (2021). The authors propose a residual U-Net for estimating S-PET images from L-PET and MRI images and return a corresponding measure of uncertainty. However, the realizations of the idea are different. While Sudarshan et al. (2021) model uncertainty in the neural network output through the per-voxel heteroscedasticity of the residuals between the predicted and the high-quality ground-truth images, we estimate it via sampling of the posterior distribution by employing the learned generator.

Our generator follows the architecture proposed in ESRGAN (Wang et al., 2018a) and Real-ESRGAN (Wang et al., 2021), which are currently state-of-the-art cGANs for image super-resolution. To achieve stochasticity, we combine the generator with the noise-injection procedure proposed in StyleGAN (Karras et al., 2019, 2020), which is a GAN model for style-based image synthesis. The discriminator in our cGAN is a pretrained ResNet34 (He et al., 2016), which is fine-tuned during training. To the best of our knowledge, such a cGAN model was not yet employed in the PET image reconstruction. A cGAN architecture closest to ours was proposed by Man et al. (2022). The authors use slightly different RRDBs accompanied with FiLM blocks and a different noise-injection procedure. Finally, the authors use posterior sampling for JPEG image decoding with high perceptual quality.

## 3. Method

In Bayesian inversion, the unknown model parameters that we want to recover and the observations are assumed to be realizations of random variables. We assume that an unknown S-PET image $X \in \mathcal{X}$ is a random vector with density $p_X$. Furthermore, we assume L-PET and T1 MRI images are an observation denoted by a random vector $Y \in \mathcal{Y}$. Our goal is to learn a generator $G_\theta(Y, Z)$ that provides an estimate $\hat{X}$ of $X$ given $Y$ via posterior sampling. Namely, $G_\theta(Y, Z)$ is a deep neural network with parameters $\theta$ that we use to approximate the posterior $p_{X|Y}$, and $Z \sim \mathcal{N}(\mathbf{0}, \mathbf{I})$ is a random vector that enables stochasticity. Sampling

from the posterior provide many S-PET estimates given the same L-PET images which allows us to quantify uncertainty.

To achieve this goal, we use a cGAN whose generator is conditioned on the two-channel input $Y$ and generates high-quality outputs consistent with the observations. Our training procedure consists of minimizing a loss function that consists of several terms. First, we use an adversarial loss term (Goodfellow et al., 2014)

$$\mathcal{L}_{adv}(G_\theta, D_\phi) = \mathbb{E}_X[\log D_\phi(X)] + \mathbb{E}_{Y,Z}[\log(1 - D_\phi(G_\theta(Y, Z)))], \tag{2}$$

where $D_\phi$ is a discriminator with parameters $\phi$. To stabilize cGAN training, in addition to the adversarial loss, we penalize the discriminator's gradients on the true data distribution (Mescheder et al., 2018), leading to the regularization term

$$\mathcal{L}_{grad}(D_\phi) = \frac{\gamma}{2}\mathbb{E}_X[||D_\phi(X)||^2]. \tag{3}$$

Since the L-PET image is obtained from a sinogram that represents raw measurements obtained by the detectors, we introduce a loss term that makes the output of the generator consistent with the observations. Let us denote the Radon operator with $\mathcal{R}$ and the L-PET image with $Y_L$, then the consistency loss can be given by

$$\mathcal{L}_c(G_\theta) = \mathbb{E}_{Y,Z}[||\mathcal{R}(Y_L) - \mathcal{R}(G_\theta(Y, Z))||_2^2]. \tag{4}$$

However, there are many plausible solutions that correspond to the measurements within the noise level, i.e., a variety of S-PET images correspond to the same L-PET image. In opposition to most of the related work, our stochastic method based on the sampling from the posterior is capable of providing a variety of plausible S-PET samples given the same L-PET image.

Training GANs is often concerned with mode collapse – a failure resulting in a GAN producing a small set of similar outputs over and over again. As we only have one S-PET per given L-PET image in the dataset, we noticed that training the GAN with the aforementioned losses results in mode collapse. Even though we expect a variety of outputs given the same L-PET image, the generator almost completely ignores the random vector $Z$ and returns a single output. To avoid this failure, we incorporate a simple regularization on the generator similar to a term proposed in (Mao et al., 2019) and (Yang et al., 2019):

$$\mathcal{L}_d(G_\theta) = \mathbb{E}_{Y,Z_1,Z_2}[||G_\theta(Y, Z_1) - G_\theta(Y, Z_2)||_1]. \tag{5}$$

By regularizing generator to maximize this term, we directly penalize the mode-collapse behaviour and force it to generate diverse outputs. Finally, we add a first-moment penalty term, proposed by Ohayon et al. (2021):

$$\mathcal{L}_{fm}(G_\theta) = \mathbb{E}_{X,Y}[||X - \mathbb{E}_Z[G_\theta(Y, Z)|Y]||_2^2], \tag{6}$$

which specifies that the expectation of many $G_\theta(Y, Z)$ for different $Z$ given the same $Y$ should be close to the ground truth $X$. As reported in (Ohayon et al., 2021) and (Man et al., 2022), Equation (6) does not limit the perceptual quality of the generated samples and further strengthens the overall optimization.

Our full objective function is given by

$$\min_{\theta} \max_{\phi} \mathcal{L}_{adv}(G_\theta, D_\phi) - \lambda_{grad}\mathcal{L}_{grad}(D_\phi) + \lambda_c\mathcal{L}_c(G_\theta) + \lambda_d\frac{1}{\mathcal{L}_d(G_\theta) + \tau} + \lambda_{fm}\mathcal{L}_{fm}(G_\theta), \quad (7)$$

where $\tau$ is a small constant for numerical stability. We solve Equation (7) in a two-step training process – while the generator is fixed, we update the discriminator and vice versa.

To sample from the posterior, we inject noise $Z$ in a StyleGAN-like fashion (Karras et al., 2019, 2020). To provide the generator's outputs with stochastic details, we feed a dedicated noise image to each RRDB. The single-channel noise images are comprised of uncorrelated Gaussian noise. The rationale behind this is that we do not want to use network capacity to implement stochastic variation as traditional generators do. This way the network does not need to invent spatially-varying stochastic details from earlier activations, but since a dedicated set of per-pixel noise is available for every RRDB, it becomes a local problem. That way the global effects are controlled by the input tensor $Y$, and the noise affects only stochastic variation. Please refer to Appendix A for more details on the architecture.

## 4. Experiments

### 4.1. Datasets

We conducted experiments on the widely used and publicly available BrainWeb dataset (Cocosco et al., 1997). It consists of MRI slices of 20 simulated brain volumes. Ground-truth synthetic PET activity was simulated using the BrainWeb library (da Costa-Luis, 2020). The S-PET reconstructions were simulated with a projector that approximates the geometry and resolution of the Siemens Biograph mMR without incorporating noise and errors caused by the wrong detection. We simulated L-PET and very-low-dose PET (vL-PET) reconstructions using a model of a truncated PET system whose details are given in Appendix B. We used 18 randomly picked brain volumes for training and 2 for testing. For every brain, we had 3 different simulations of PET activity. Each MRI volume in the dataset is of $256 \times 256 \times 258$ dimensions. We have taken only slices with notable PET activity, i.e., the slices that approximately correspond to the whole brains, leaving us with $256 \times 256 \times 132$ volume grid. Thus, the two BrainWeb training datasets consisted of 7,128 L-PET or vL-PET and S-PET slices, and for testing we used the remaining 792 slices.

Additionally, we conducted experiments on a real-world dataset from the Alzheimer's disease neuroimaging initiative (ADNI) database. The slices from the ADNI database were used as S-PET ground truths and vL-PET reconstructions were simulated similarly as in the BrainWeb dataset (see Appendix B for more details.) A total of 9 subjects were selected. For each subject, we were provided with six different brain T1 MRI and PET scans. The MRI and PET slices were obtained separately so we registered them before training. Each brain scan in the dataset consists of 60 transaxial slices. The dimension of each slice is $128 \times 128$ pixels. The ADNI training dataset consisted of 2,880 vL-PET and S-PET slices, and the test dataset consisted of 360 slices.

### 4.2. Experimental Results

We concatenated the L-PET and high-quality MRI images into two-channel $m \times n$ tensors and used them as an input $Y$ that conditions our cGAN model. Every RRDB was supplied

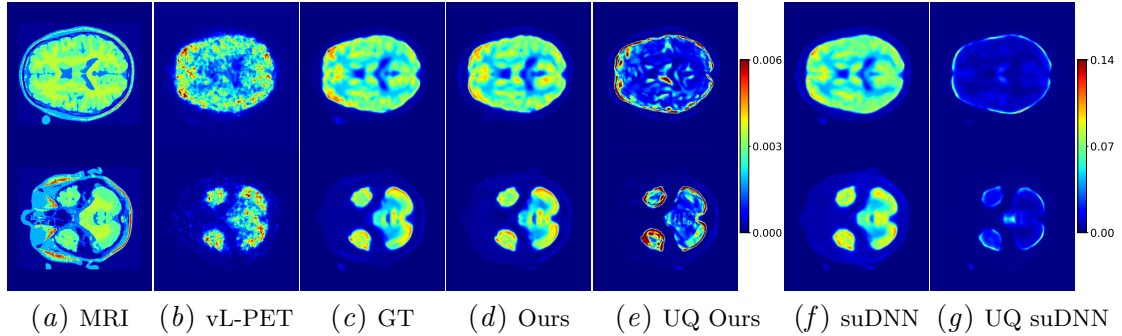

$(a)$ MRI    $(b)$ vL-PET    $(c)$ GT    $(d)$ Ours    $(e)$ UQ Ours    $(f)$ suDNN    $(g)$ UQ suDNN

Figure 2: Reconstruction examples for the BrainWeb dataset. (a) and (b) MRI and vL-PET inputs that condition the generator; (c) S-PET ground truths; (d) and (f) our and suDNN (Sudarshan et al., 2021) S-PET reconstructions obtained as the mean of 24 posterior samples and the mean of 24 different suDNN outputs, respectively; (e) and (g) our and suDNN uncertainty estimates representing the variance of 24 posterior samples and the mean of 24 different suDNN variance estimates.

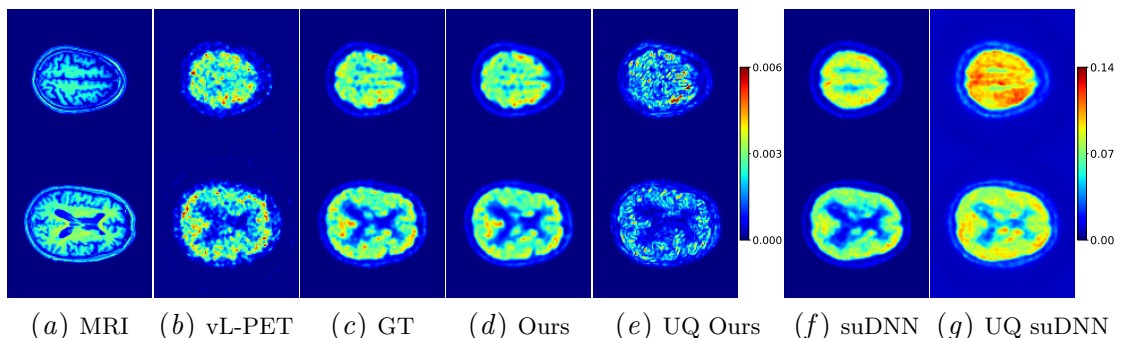

$(a)$ MRI    $(b)$ vL-PET    $(c)$ GT    $(d)$ Ours    $(e)$ UQ Ours    $(f)$ suDNN    $(g)$ UQ suDNN

Figure 3: Reconstruction examples for the ADNI dataset. Column representation is the same as in Figure 2.

with a fresh noise image that promotes stochastic variation in the generator's output. Please refer to Appendix A for the training details.

Figure 2 shows results obtained by using the proposed method on the BrainWeb dataset for vL-PET input slices. We compare our method with the suDNN framework. Both methods enhance the vL-PET image reconstructions. Our generator outputs visually better and more accurate reconstructions in comparison to the S-PET reconstructions obtained by suDNN. While our measure of uncertainty is obtained by calculating the variance of the posterior samples, in the suDNN framework, it is estimated by the model and is one of the network's outputs. We believe our method outputs a measure of uncertainty which is more physically-meaningful and interpretable than the suDNN's UQ. Additionally, notice that for the vL-PET setting (Figure 2) our method provides maps with higher uncertainty than

Table 1: Reconstruction results in terms of PSNR in dB and SSIM on the BrainWeb dataset.

|  |  | MLEM | suDNN | Ours |
|---|---|---|---|---|
| **L-PET** | PSNR | 29.18 | 31.11 | **37.45** |
|  | SSIM | 0.8205 | 0.9364 | **0.9746** |
| **vL-PET** | PSNR | 22.92 | 28.30 | **32.15** |
|  | SSIM | 0.4785 | 0.9170 | **0.9517** |

Table 2: Reconstruction results in terms of PSNR in dB and SSIM in on the ADNI dataset.

|  |  | MLEM | suDNN | Ours |
|---|---|---|---|---|
| **vL-PET** | PSNR | 24.40 | 25.91 | **31.97** |
|  | SSIM | 0.5197 | 0.8154 | **0.9216** |

for the L-PET setting (Figure 1). It is to be expected since the L-PET reconstructions in Figure 1 are obtained from much more coincidences (measurements) than the vL-PET reconstructions. In contrast, the suDNN framework does not provide such a behavior.

Figure 3 shows the comparison between our and suDNN reconstruction results and UQs for the real-world ADNI dataset for vL-PET inputs. Our method again outperforms suDNN in the reconstruction quality and meaningfulness of the uncertainty maps.

We calculate the uncertainty as the variance of 24 randomly picked posterior samples for the same input $Y$. While the inputs and the PET reconstructions were scaled to $[0, 1]$, the variance was scaled so that the UQ results in the last column of Figure 2 are visually satisfactory. We provide additional experimental results in Appendix C, where we show that our framework yields diverse and high-quality posterior samples.

In Table 1, we provide a quantitative measure of reconstruction results for the BrainWeb dataset in comparison to the MLEM reconstruction algorithm, which is a golden standard in PET image reconstruction, and the suDNN framework. The results are given in terms of the peak signal-to-noise ratio (PSNR) in decibels and the structural similarity index measure (SSIM). The results in the table are the means of all the slices of interests in the testing brains. For calculating the PSNR and SSIM of the reconstructions obtained by our method, as a reference S-PET estimation we used the mean of 24 generated posterior samples. In Table 2, we show a similar comparison of the reconstruction results for the ADNI dataset. For both datasets, our framework yields better reconstruction results than the suDNN framework. We provide the ablation study results in Appendix D.

Our framework yields high-quality reconstruction results for synthetic and real-world datasets that were used for training, however it is yet to be seen how it behaves on out-of-distribution data. Additionally, since the model is trained on the brain datasets, we can expect some difficulties when used beyond brain. Even though we believe we can extend the framework and train it on other body parts or even the whole body, it is not clear how well can this be translated and can the model generalize well. We leave this for future research.

## 5. Conclusion

We proposed a deep learning-based framework for uncertainty quantification in PET image reconstruction via posterior sampling. The framework estimates a standard-dose PET reconstruction from a low-dose PET reconstruction and a high-quality MRI image. The estimated standard-dose PET image is provided with a corresponding measure of uncertainty which is calculated as the variance of different posterior samples. We demonstrated that the framework yields high-quality reconstructions that are consistent with the measurements and physically-meaningful quantification of uncertainty. The proposed framework can have a great potential in clinics by providing a more reliable interpretation of PET image reconstructions, and thus helping the medical practitioners in making critical decisions.

## Acknowledgments

Data used in the preparation of this article were obtained from the Alzheimer's Disease Neuroimaging Initiative (ADNI) database (`adni.loni.usc.edu`).

The authors thank Uddeshya Upadhyay for constructive discussion on his work (Sudarshan et al., 2021).

The authors gratefully acknowledge financial support from the Croatian Science Foundation under Projects IP-2019-04-6703 and IP-2019-04-4189.

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

## Appendix A. Implementation Details

### A.1. Architecture

Our generator's architecture is illustrated in Figure 4. We use pixel unshuffling to rearrange a large-scale $m \times n \times c$ input image into a $m/s \times n/s \times s^2 c$ input tensor. It significantly

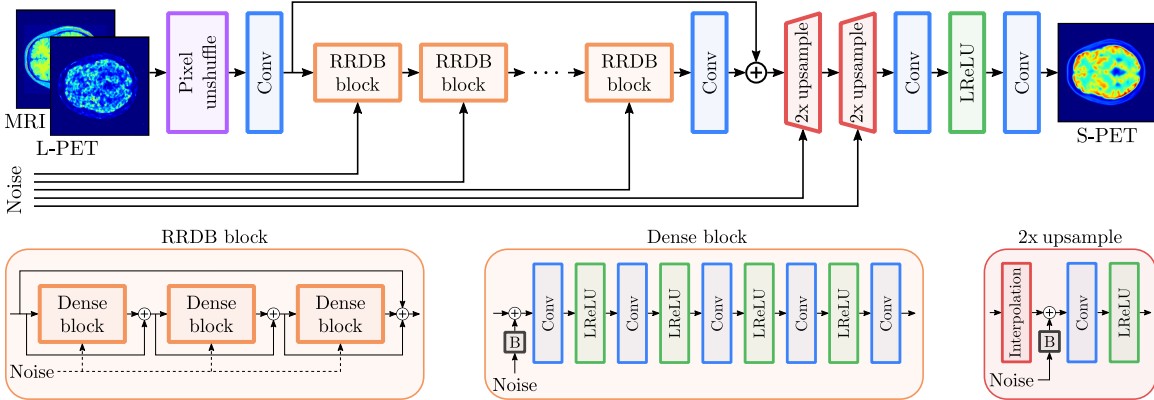

Figure 4: The generator's architecture is based on the architectures of ESRGAN (Wang et al., 2018a, 2021) and StyleGAN (Karras et al., 2019, 2020). The two-channel input tensor of spatial size $256 \times 256$ ($128 \times 128$) is first pixel-unshuffled into a $64 \times 64 \times 32$ ($64 \times 64 \times 8$) tensor to rearrange spatial information into channels. Residual-in-residual dense blocks (RRDBs) are used to reconstruct the S-PET image from the unshuffled input, which is afterwards upsampled to the original spatial size of the input. To allow for posterior sampling, a fresh set of Gaussian noise is added at the beginning of every dense block and in the upsampling blocks. The noise input is per-channel scaled by learned factors denoted by "B" blocks.

reduces computational complexity in the following RRDBs where each input tensor of a dense block has 64 channels. The total number of RRDBs is 23. The upsampling blocks first interpolate the input tensor two times and then add a dedicated noise image scaled by learned factors. Finally, the output is a one-channel $m \times n$ S-PET image. The generator has a total of 16.7M trainable parameters. Our cGAN's discriminator is a ResNet34 (He et al., 2016).

### A.2. Training

We trained the model using the Adam optimizer (Kingma and Ba, 2014) with $\beta_1 = 0.9$ and $\beta_2 = 0.999$ on two 11GB NVIDIA RTX2080 Ti graphics cards. The model was trained alternately, by updating the discriminator while the generator was fixed and vice versa. We used the non-saturating adversarial loss (Goodfellow et al., 2014) for training the cGAN. The batch-size was set to 4.

For the L-PET BrainWeb dataset, we trained the model for 50 epochs. The learning rate was set to $2 \times 10^{-4}$. Other regularization parameters were set to: $\lambda_{grad} = 0.6$ and $\gamma = 10$, $\lambda_c = 5 \times 10^{-4}$, $\lambda_d = 2 \times 10^{-4}$ and $\tau = 1 \times 10^{-5}$, and $\lambda_{fm} = 2$.

For the vL-PET BrainWeb dataset, we trained the model for 100 epochs. The learning rate was set to $1 \times 10^{-4}$. Other regularization parameters were set to: $\lambda_{grad} = 0.6$ and $\gamma = 10$, $\lambda_c = 3 \times 10^{-4}$, $\lambda_d = 1 \times 10^{-4}$ and $\tau = 1 \times 10^{-5}$, and $\lambda_{fm} = 2$.

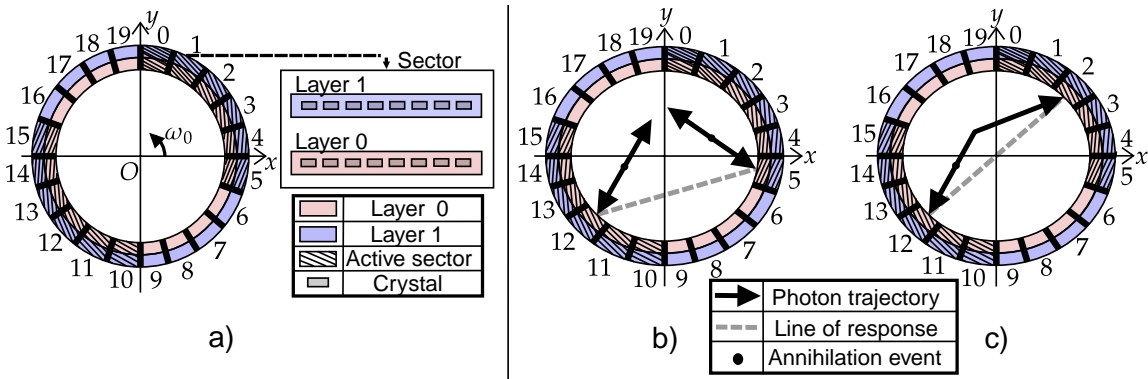

Figure 5: a) Virtual 2D PET system, b) Random coincidence, c) Scattered coincidence.

For the vL-PET ADNI dataset, we trained the model for 150 epochs. The learning rate was set to $1 \times 10^{-4}$. Other regularization parameters were set to: $\lambda_{grad} = 0.6$ and $\gamma = 10$, $\lambda_c = 3 \times 10^{-4}$, $\lambda_d = 1 \times 10^{-4}$ and $\tau = 1 \times 10^{-5}$, and $\lambda_{fm} = 2$.

To calculate the diversity loss in Equation (5) and the mean of generated samples in Equation (6), we generate 6 different outputs using different realizations of noise for each of the first 2 input images in the batch.

### A.3. SuDNN Architecture and Training

We trained the suDNN model as proposed in (Sudarshan et al., 2021). For training, we used multimodal 2.5D input data consisting of L-PET or vL-PET and T1 MRI slices. The residual U-Net architecture was composed of the same number of layers and skip connections as proposed in (Sudarshan et al., 2021). The values of two constants that were not reported in (Sudarshan et al., 2021) were set to $\tau = 1 \times 10^{-5}$ – a small constant for numerical stability, and regularization parameter $\lambda = 1 \times 10^{-4}$, $3 \times 10^{-4}$, and $5 \times 10^{-4}$, for the L-PET BrainWeb dataset, vL-PET BrainWeb dataset and vL-PET ADNI dataset, respectively. We trained the suDNN model for 500 epochs for both L-PET and vL-PET BrainWeb datasets and 1300 epochs for the ADNI dataset.

## Appendix B. PET Scan Simulation

To simulate a realistic PET measurements, a virtual 2D PET system based on the ClearPET system (Ziemons et al., 2005) was implemented. The entire PET system is depicted in Figure 5. The ClearPET system consists of 20 sectors that are placed uniformly around the circle. Each sector has two layers and each layer is a grid of 8 scintillation crystals, i.e. detectors. The imaging system rotates with constant angular velocity $\omega_0$. During one measurement, the system makes a full circle.

Instead of using a fully equipped PET system, we take a partially equipped system; only 12 sectors are active, 6 on each side of a circle. In such a system, the probability of detecting an event changes with its position (Matulić et al., 2021). Thus, filtered-back projection is not a suitable reconstruction method. Introducing the system matrix, the core of the

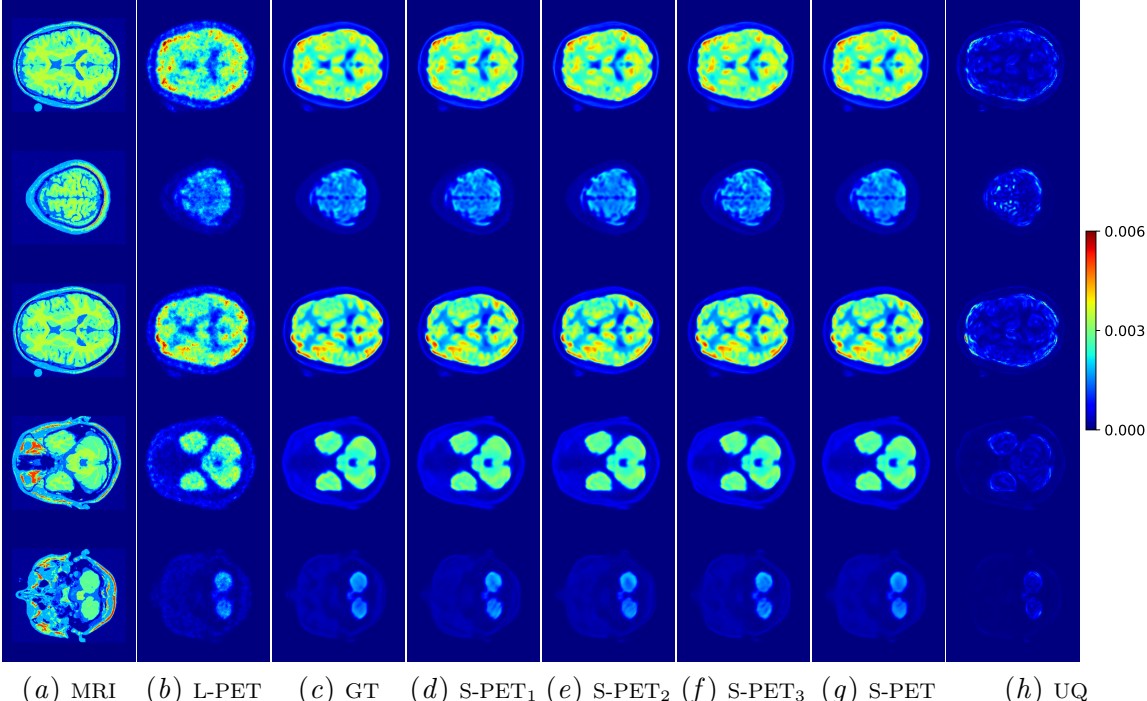

$(a)$ MRI $\quad$ $(b)$ L-PET $\quad$ $(c)$ GT $\quad$ $(d)$ S-PET$_1$ $(e)$ S-PET$_2$ $(f)$ S-PET$_3$ $(g)$ S-PET $\quad$ $(h)$ UQ

Figure 6: Reconstruction examples for the BrainWeb dataset. In (a) and (b) columns there are inputs, MRI and L-PET, that condition the generator. In (c) column there are S-PET ground truths. In (d), (e) and (f) columns there are examples of posterior samples for different noise realizations $Z$. In column (g) there are means of 24 posterior samples and in column (h) there are corresponding uncertainty estimates.

MLEM reconstruction algorithm, gives us the ability to perform a mapping between pixels and detectors. As a consequence, the MLEM algorithm doesn't suffer from the probabilistic nature of PET systems, making it a golden standard in PET image reconstruction.

To simulate the PET measurement accurately, we incorporated three effects that occur during the measurement: non-zero momentum of electron and positron, random and scattered coincidences.

The first effect is a result of electron-positron annihilation. If the momentum of the system containing both particles is zero, then an angle between two created photons is exactly $180°$ due to the conservation of momentum. But in the real world, the momentum is not equal to zero. Thus, the angle between two photons is in a range between $180° \pm 0.5°$.

Random coincidences arise when two photons from different annihilation processes hit detectors in a short time-window, creating a false event, as shown in Figure 5.

Scattered coincidences happen when at least one of the photons is scattered in its trajectory toward the detector, as depicted in Figure 5. We have modeled scattered coincidences in the following way – only one photon undergoes a single scattering. A

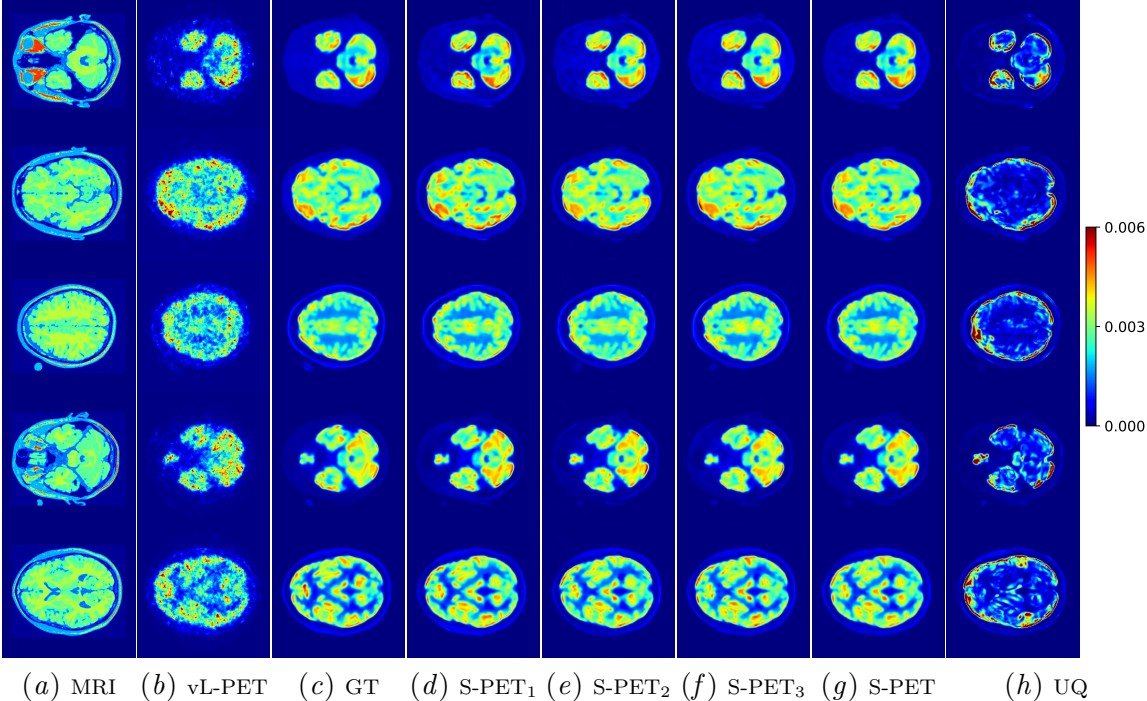

$(a)$ MRI   $(b)$ vL-PET   $(c)$ GT   $(d)$ S-PET$_1$ $(e)$ S-PET$_2$ $(f)$ S-PET$_3$ $(g)$ S-PET   $(h)$ UQ

Figure 7: Reconstruction examples for the BrainWeb dataset. In (a) and (b) columns there are inputs, MRI and vL-PET, that condition the generator. In (c) column there are S-PET ground truths. In (d), (e) and (f) columns there are examples of posterior samples for different noise realizations $Z$. In column (g) there are means of 24 posterior samples and in column (h) there are corresponding uncertainty estimates.

deflection angle follows the Henyey-Greenstein phase function (Toublanc, 1996)

$$P_{HG}(\theta, g) = \frac{1 - g^2}{(1 + g^2 - 2g\cos(\theta))^{\frac{3}{2}}},$$

where $g$ is the asymmetry factor. The asymmetry factor is set to 0.98, a value associated with biological tissue (Binzoni et al., 2006).

The simulation of the entire brain is done slice by slice. The total number of coincidences generated for one slice is proportional to a ratio between the activity of that slice and the whole brain. The effect of the non-zero momentum of the electron and positron is included in all coincidences. Also, we can tune a number of random and scattered coincidences.

Each brain in the BrainWeb dataset is simulated with described setup. We conducted two experiments on the BrainWeb dataset - low-dose PET (L-PET) simulation and very low-dose PET (vL-PET) simulation. During the low-dose PET simulation, we generated $N_{total} \approx 25$M coincidences throughout the entire brain of which $p_r = 15\%$ were random, and $p_s = 15\%$ were scattered coincidences. Each L-PET slice was reconstructed by the MLEM

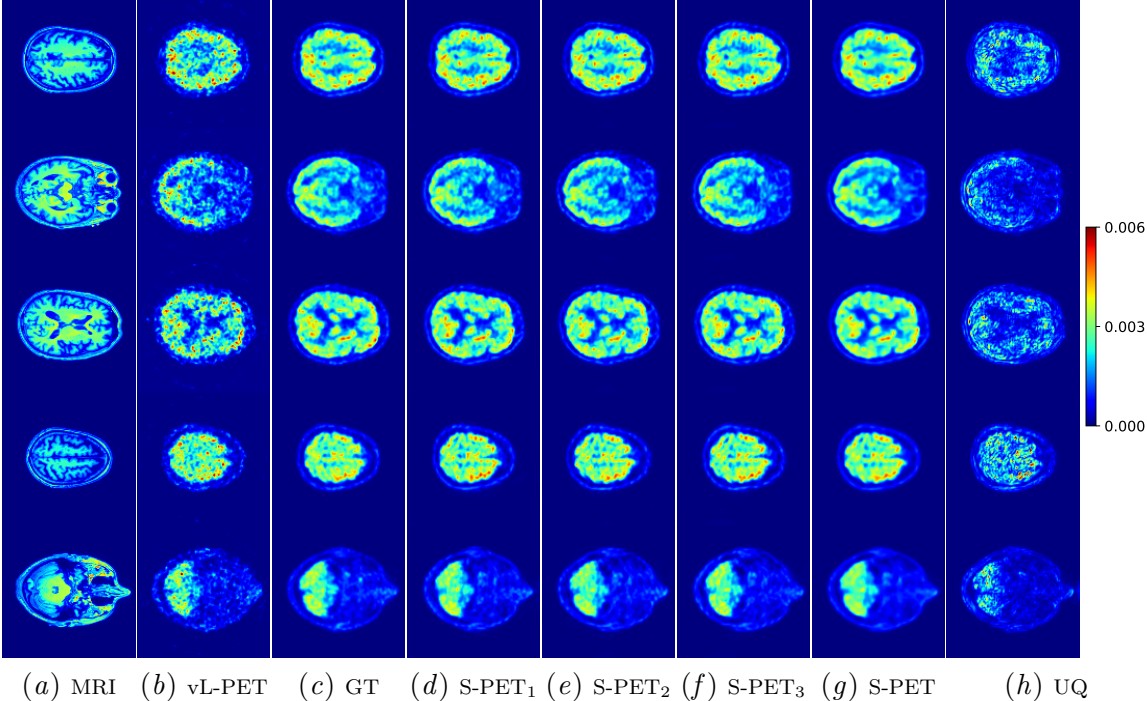

$(a)$ MRI  $(b)$ vL-PET  $(c)$ GT  $(d)$ S-PET$_1$ $(e)$ S-PET$_2$ $(f)$ S-PET$_3$ $(g)$ S-PET  $(h)$ UQ

Figure 8: Reconstruction examples for the ADNI dataset. In (a) and (b) columns there are inputs that condition the generator. In (c) column there are S-PET ground truths. In (d), (e) and (f) columns there are examples of posterior samples for different noise realizations $Z$. In column (g) there are means of 24 posterior samples and in column (h) there are corresponding uncertainty estimates.

algorithm. Similarly, during the very low-dose PET simulation, we generated $N_{total} \approx$ 5M coincidences throughout the entire brain. We kept the same parameters and reconstruction method as in low-dose PET simulation.

We were guided by the same principle for slices gathered from the ADNI dataset. To simulate a very low-dose PET, we generated $N_{total} \approx$ 2M coincidences throughout the entire brain. We kept the same contribution of random and scattered coincidences as in the BrainWeb dataset. Also, we used the MLEM reconstitution algorithm.

## Appendix C. Additional Results

We provide additional results for three different settings. We show various reconstruction results for the same input corresponding to posterior samples obtained using our framework. Furthermore, we provide the mean of 24 posterior samples for different noise realizations $Z$ while given the same input $Y$. In Figure 6, we show reconstruction results and the corresponding measure of uncertainty for L-PET simulations. In Figure 7, we provide examples of reconstructions for vL-PET inputs. Notice that in Figure 7 in comparison to

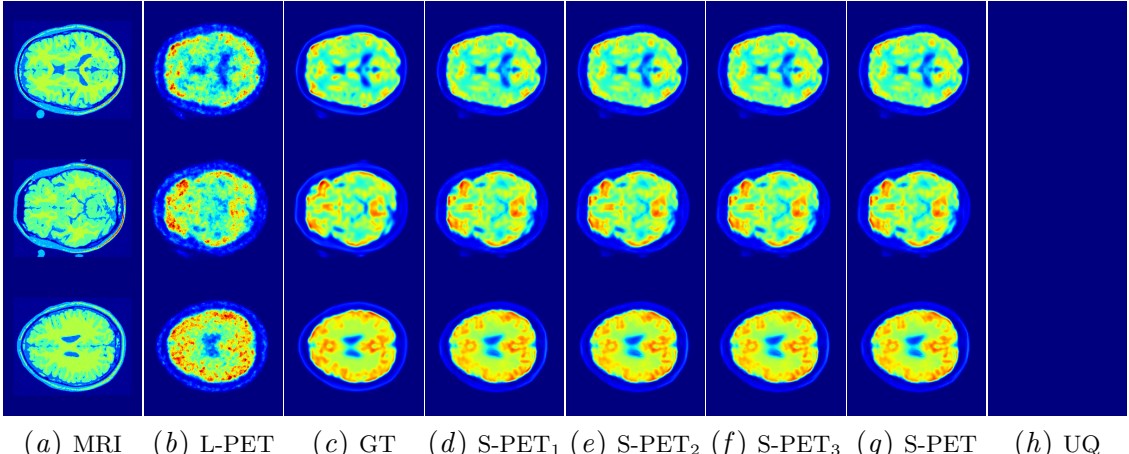

$(a)$ MRI    $(b)$ L-PET    $(c)$ GT    $(d)$ S-PET$_1$ $(e)$ S-PET$_2$ $(f)$ S-PET$_3$ $(g)$ S-PET    $(h)$ UQ

Figure 9: Ablation study reconstruction results. In (a) and (b) columns there are inputs that condition the generator. In (c) column there are S-PET ground truths. In (d), (e) and (f) columns there are examples of posterior samples for different noise realizations $Z$. In column (g) there are means of 24 posterior samples and in column (h) there are no visible corresponding uncertainty estimates.

Figure 6, the estimated uncertainty is higher. Again, such results are expected since L-PET input images are of higher quality compared to vL-PET images and are obtained from much more coincidences. Furthermore, the L-PET inputs lead to smaller errors between estimated S-PET reconstructions and the ground truths than the vL-PET inputs, and thus reconstructions from L-PET are far less uncertain. In Figure 8, we show some of the additional reconstructions for the real-world ADNI dataset for vL-PET inputs. We show three randomly picked posterior samples, the mean of 24 different posterior samples and the measure of uncertainty.

Same as in the main part of the paper, we scaled PET reconstructions to $[0, 1]$ and uncertainty maps to $[0, 0.006]$ so that the results for different datasets can be compared.

## Appendix D. Ablation Study

We removed the diversity loss term Equation (5) and the first-moment penalty term Equation (6) from the full objective function Equation (7) and trained the cGAN with the same training parameters and number of epochs. Reconstruction results are given in Figure 9. We observe mode-collapse behaviour – there is no diversity in posterior samples as the network almost completely ignores noise $Z$. As a consequence, we are unable to estimate uncertainty.

While the reconstruction quality is again better than that of the MLEM reconstruction algorithm, it is a bit lower in terms of the PSNR and SSIM in comparison to the model trained using the full objective function Equation (7). The results are given in Table 3.

Table 3: Ablation study reconstruction results in terms of the PSNR in dB and the SSIM.

|  |  | MLEM | Ours-Ablated |
|---|---|---|---|
| **L-PET** | PSNR | 29.18 | 34.61 |
|  | SSIM | 0.8205 | 0.9310 |
| **vL-PET** | PSNR | 22.92 | 29.66 |
|  | SSIM | 0.4785 | 0.9097 |

