# OpenReview forum: "Estimating Uncertainty in PET Image Reconstruction via Deep Posterior Sampling"
_MIDL.io/2023/Conference — MIDL 2023 Poster_

### Official Review · Reviewer_vGgT · 2023-01-27

**Confidence:** 5
**Preliminary Rating:** 1

**Summary:**

The paper proposes to train a conditional GAN to be able to generate a distribution of high-dose PET images conditioned on low-dose PET images. The paper claims this as "posterior sampling". The methodology is standard and uses standard components. The training strategy is standard. The results are on a small dataset.

**Strengths:**


Uncertainty modeling in medical imaging is important.

The authors leverage the current literature on GANs towards solving their problem. The framing of the reconstruction problem uses a consistency term using Radon transforms on the cGAN output.



**Weaknesses:**


The baseline used is only MLEM that is a very old baseline. Newer methods need to be compared in Table 1.

There is little methodological novelty in the paper.

The empirical analysis is only using a simulate dataset.


**Deanonymize Review:**

no

**Detailed Comments:**


Please see the comments above.

**Paper Type:**

both

**Questions To Address In The Rebuttal:**


Limited methodological novelty.

The paper guises the problem using Bayesian terms, but this is really a conditional GAN using a consistency term on the generated outputs.

Lack of evaluation on a real-world dataset.

Lack of evaluation on recent relevant baselines.

---

### Official Review · Reviewer_GeVb · 2023-02-02

**Confidence:** 3
**Preliminary Rating:** 4
**Recommendation:** Poster

**Summary:**

The manuscript "Estimating Uncertainty in PET Image Reconstruction via Deep Posterior Sampling" proposes a deep learning (DL)-based approach to reconstruct standard-dose Positron Emission Tomography (PET) images from low-dose PET and magnetic resonance images (MRI) and simultaneously provide an uncertainty quantification (UQ).
The abstracts summarized the key aspects of the work and selected keywords represent the content of the manuscript.
The introduction describes the problem in sufficient detail and motivates the scientific questions well. State-of-the-art methods are described in sufficient detai. Related and recent work is introduced and context to the presented work is given to guide the reader. The methods used in this work are explained in a comprehensive fashion in the methods section. The annex refers to some technological details to not descruct the reader in this section. The experimental setup and underlying data are well described. The results section summarizes the results rather short before the manuscirpt ends with a short concluding statement.
In my opinion the most relevant part of the work is the uncertainty estimation while the transformation tasks (L-PET + MRI -> S-PET) have been publised elsewere already (references to existing publications in this field are given in the manuscrpt and situation is clearly named).

Update after Review 1: Changed rating from "borderline" to "weak accept"

**Strengths:**

The work addresses a very important aspect of modern data-driven approaches, namely the estimation and/or quantification of the uncertainty of the network's prediction. This is also very relevant in the field of image reconstruction and in particular for medical applications. Tackling this problem will help to increase acceptance of data-driven methods in the field of medical imaging and advanced data processing.
The authors give a overview of related work and reference many relevant publications.

**Weaknesses:**

While the authors explain in sufficient detail the problem and their motivation, the results are rather shortly reported, i.e. only two global figures (PSNR, SSIM) are given without any error/uncertainty measure. Little effort was made to describe how the results were evaluated (e.g. best / worst test case, variablity of data) and derived uncertainty, which is one main topic of this work, was not quantified. The presented work is limited to simulated data and no test/validation was performed on real-world data.
In my opinion, the conclusion is also rather brief and does not mention any limitations, possible pitfalls etc. Furthermore, the results are not discussed which I would consider as one of the most important aspects of scientific publications.

**Deanonymize Review:**

no

**Detailed Comments:**

- Fig. 2: unknown scaling/windowing for the subfigures. Therefore, it is difficult to estimate the level of UQ from the maps alone.

**Paper Type:**

methodological development

**Questions To Address In The Rebuttal:**

- Did you test your network on real data? If so, how was the performance?
- Have you quantified the uncertainty (maps)?
- I would kindly invite you to broadly discuss your work and the expected limitations. The given conculsion is rather short and not very balanced / critical.

---

### Official Review · Reviewer_9V19 · 2023-02-03

**Confidence:** 4
**Preliminary Rating:** 4
**Recommendation:** Oral, Poster

**Summary:**

This work presents a conditional GAN-based reconstruction approach for PET imaging. The architectures is mostly based on state of the art superresolution models, but it is extended with hand-picked aspects from other GAN approaches, in particular for noise injection, in order to let the generator produce multiple plausible high-quality reconstructions. The authors use simulated low-dose PET images based on the BrainWeb database and related software, so that the ground-truth standard-dose PET reconstruction is known. The results show high-quality reconstructions and "physically plausible" uncertainty maps.

**Strengths:**

The method is a well-motivated combination of state-of-the art components and at least one important ablation study is given in the appendix. The results show a clear improvement in the PET reconstruction quality on the BrainWeb dataset, which could be a clinically relevant contribution. The fact that simulated data was used for training and evaluation leads to credible results on this dataset. The simulation seems to be involved and physically plausible (but is only described in the appendix).

**Weaknesses:**

This work is more like a first feasability study and cannot claim clinical applicability yet, which is also the reason why I marked it as a method paper. I come to this conclusion because the reconstruction has only been done on simulated images derived from the BrainWeb database, and it is not clear how well this can be translated to other body regions and more diverse datasets.

The method was trained with only 7k 2D image slices, and tested on about 800 ones.  A validation set is not mentioned, but the method has a number of hyperparameters (the most important of which seem to be given in the appendix), so it would be interesting to learn how these were set, and how robust the results are.

Some important parts of the paper are in the appendix.  I have looked at them out of interest, but for fairness, I will focus on the main part of the manuscript (and also give no detailed feedback on the appendix below).

**Deanonymize Review:**

yes

**Detailed Comments:**

The paper mentions a *lot* of (seemingly) relevant literature. I was unsure whether I should put this as a strength, but one could also see it as a weakness in the sense that it makes it harder for the reader to understand what are the most important references for this paper, which actually includes a few rather general introductions into inverse problems, Bayesian methods, and PET imaging that I would not have expected in this depth in a paper of this length.

"7, 128 L-PET images" should be written without space (and you might want to use the term "slice" to be more specific and unambiguous).



**Paper Type:**

methodological development

**Questions To Address In The Rebuttal:**

I would like the authors to comment on my above questions about the hyperparameter tuning and the missing validation set.

Second, I would like to know why "only slices with notable activity" have been used.  Wouldn't including them have been useful to show that the reconstruction also works for those slices?  From an application perspective, I can see that to be an important fact to check.  In fact, one could still have looked at it separately, but I would have really liked to see quantitative results on these slices as well in table 1.

Maybe you could also give an outlook on the possibilty to extend this work beyond the brain, e.g., to oncological whole-body scans.

I noticed that the consistency loss uses a Radon operator on the low-dose PET image and it surprised me because I thought the sinograms could have been part of the simulation. But maybe that's what is being done and the mathematical formulation of the loss function just gives the false impression that it is computed later. Out of interest, I would like a comment on that – it is not something that would change my mind w.r.t. the paper.

---

### Meta-Review · Area_Chair_YZmQ · 2023-02-24

**Recommendation:** Accept (Poster)
**Confidence:** 4

**Metareview:**

Even though the reviewers had initial concerns regarding a lack of novelty of the approach and an evaluation limited to simulated data, the authors adequately addressed their comments by clarifying the methodological novelty and adding experiments with real data, on top of other substantial edits made to the manuscript. All reviewers highlight the importance of the topic tackled, uncertainty estimation. Overall, the paper would be a good MIDL contribution.